# The indirect impact of COVID-19 pandemic on inpatient admissions in 204 Kenyan hospitals: An interrupted time series analysis

Steven Wambua[1]*, Lucas Malla[2], George Mbevi[2], Amen-Patrick Nwosu[3], Timothy Tuti[2], Chris Paton[3], Samuel Cheburet[4], Ayub Manya[4], Mike English[2,3‡], Emelda A. Okiro[1,3‡]

**1** Population Health Unit, Kenya Medical Research Institute-Wellcome Trust Research Programme, Nairobi, Kenya, **2** Health Services Unit, KEMRI-Wellcome Trust Research Programme, Nairobi, Kenya, **3** Oxford Centre for Global Health Research, Nuffield Department of Clinical Medicine, University of Oxford, Oxford, United Kingdom, **4** Ministry of Health, Nairobi, Kenya

‡ Joint senior authors and contributed equally.
* swambua@kemri-wellcome.org

**Data Availability Statement:** Aggregated DHIS2 data is available online with access provided by

## Abstract

The first case of severe acute respiratory coronavirus 2 (SARS-CoV-2) was identified in March 2020 in Kenya resulting in the implementation of public health measures (PHM) to prevent large-scale epidemics. We aimed to quantify the impact of COVID-19 confinement measures on access to inpatient services using data from 204 Kenyan hospitals. Data on monthly admissions and deliveries from the District Health Information Software version 2 (DHIS 2) were extracted for the period January 2018 to March 2021 stratified by hospital ownership (public or private) and adjusting for missing data using multiple imputation (MI). We used the COVID-19 event as a natural experiment to examine the impact of COVID-19 and associated PHM on use of health services by hospital ownership. We estimated the impact of COVID-19 using two approaches; Statistical process control (SPC) charts to visualize and detect changes and Interrupted time series (ITS) analysis using negative-binomial segmented regression models to quantify the changes after March 2020. Sensitivity analysis was undertaken to test robustness of estimates using Generalised Estimating Equations (GEE) and impact of national health workers strike on observed trends. SPC charts showed reductions in most inpatient services starting April 2020. ITS modelling showed significant drops in April 2020 in monthly volumes of live-births (11%), over-fives admissions for medical (29%) and surgical care (25%) with the greatest declines in the under-five's admissions (59%) in public hospitals. Similar declines were apparent in private hospitals. Health worker strikes had a significant impact on post-COVID-19 trends for total deliveries, live-births and caesarean section rate in private hospitals. COVID-19 has disrupted utilization of inpatient services in Kenyan hospitals. This might have increased avoidable morbidity and mortality due to non-COVID-19-related illnesses. The declines have been sustained. Recent data suggests a reversal in trends with services appearing to be going back to pre- COVID levels.

Ministry of Health https://hiskenya.org/dhis-web-commons/security/login.action. The datasets used and/or analysed during the current study are available to others from this source on reasonable request.

**Funding:** This research was funded in whole or in part by the Wellcome Trust Intermediate Fellow [Grant No. 201866]. For the purpose of Open Access, the author has applied a CC-BY public copyright licence to any author accepted manuscript version arising from this submission. Steven Wambua and Emelda A. Okiro are supported through a Wellcome Trust Intermediate Fellow [Grant No. 201866]. Lucas Malla, Amen-Patrick Nwosu, George Mbevi, Timothy Tuti and Chris Paton are supported through Funds from the Wellcome Trust [Grant No. 207522] awarded to Prof. Mike English as a senior Fellowship together with additional funds from a Wellcome Trust core grant awarded to the KEMRI-Wellcome Trust Research Programme [Grant No. 092654]. Steven Wambua, Lucas Malla, George Mbevi, Timothy Tuti, Mike English and Emelda A. Okiro, acknowledge the support of the Wellcome Trust to the Kenya Major Overseas Programme [Grant No. 203077]. The funders had no role in study design, data collection and analysis, decision to publish, or preparation of the manuscript.

**Competing interests:** The authors declare that they have no competing interests.

# Introduction

The global impact of the severe acute respiratory coronavirus 2 (SARS-CoV-2) virus has been extensive with over 137.5 million confirmed cases and 2.9 million official deaths globally as of April 13, 2021 [1]. In Kenya, the first case of COVID-19 was confirmed on 13th March 2020. Since then, 147147 cases have been reported with 2394 official deaths confirmed by April 13, 2021 [1] even though the real extent of spread is estimated to be greater [2]. The spread of the pandemic has placed unprecedented challenge on health systems.

There have been three waves of the pandemic in Kenya. The initial stage of the outbreak following the first case triggered implementation of a partial lockdown on April 6,2020, when 158 cases and 6 deaths had been reported nationally. This wave peaked in July/August 2020, and cases started to drop gradually [2]. However, two months later, the country experienced the second tide of the pandemic, which peaked in October/November 2020 [3] and could have been caused by opening of bars and restaurants and phased re-opening of universities. To curb the spread, the government instituted more restrictions which saw a dramatic drop in cases in the following four months. Nonetheless, the third wave was experienced in March 2021, during which the second partial lockdown was instituted on 27 March 2021.

Previous outbreaks, noticeably the Ebola epidemic in West Africa in 2014 have brought to light unintended effects of control measures on utilization of health services during and after an outbreak, where admissions and surgeries reduced significantly [4–8]. This may be more pronounced in countries with weak and fragile healthcare systems [9]. This effect maybe unevenly distributed with women and children more vulnerable [4] and affecting some healthcare services more than others [10].

To capture the indirect effects of COVID-19, we sought to quantify the influence of COVID-19 confinement measures by the government on admissions and deliveries to assess utilization of inpatient services using data from 204 Kenyan hospitals.

# Materials and methods

## Chronology of events

**COVID-19 preparedness measures.**   Two months before the first case of COVID-19 was reported in Kenya on 13th March 2020, the government had increased preparedness to the pandemic. These included installation of surveillance systems to detect suspected COVID-19 cases at border points, additional medical staff at international airports and ports, in-country capacity to test and isolate COVID-19 cases, sensitization of healthcare workers on dealing with COVID-19 cases and establishment of a National Emergency Response Committee.

**COVID-19 intervention measures.**   The government started introduction of interventions to combat COVID-19 spread on 13th March 2020. These included suspension of public gatherings and events, closing of schools, international travel restrictions, fumigation and disinfection of markets, closure of bars and restaurants, suspension of attendance to places of worship, limit of people attending weddings and funerals and national dust-to-dawn curfew. The month of April 2020 saw cessation of movement in and out of four counties with highest number of COVID-19 cases, restaurants were opened under strict guidelines of social distancing, handwashing and temperature checks. During the month of May 2020, cessation of movement into and out of Kenya through Tanzania and Somalia borders was affected while on 10th June 2020 the government launched home-based care for patients with COVID-19 infection. In July 2020, certain measures were relaxed; cessation of movement into the four counties was lifted, phased re-opening of places of worship and resumption of local air travel. On 1st August 2020, international air travel resumed and in September 2020, operation of bars resumed. This

was followed by phased reopening of schools and lifting of suspension on political gatherings in October 2020 and November 2020 respectively. In December 2020, there was a nationwide health care workers strike precipitated by demands for better working conditions such as provision of adequate Personal Protective Equipment (PPE), enhanced risk allowances and a health insurance cover. The strike only affected public facilities. Public and private health facilities in Kenya have separate management processes and funding structures affecting service provision hence challenges experienced in the public sector do not mirror the situation in Private facilities. These Health worker strikes were as a result of pay disputes, contracts and COVID safety concerns, including availability of personal protective equipment [11]. By early 2021, the test positivity proportion was below 5% and all schools were re-opened on January 4th. On 5th March 2021, the government launched its COVID-19 vaccination campaign and on March 12th, due to cases rising again during the third wave of the pandemic, funerals, weddings and other events were capped to including 100 persons and capacity of places worship was restricted. The timeline of events is presented in Fig 1.

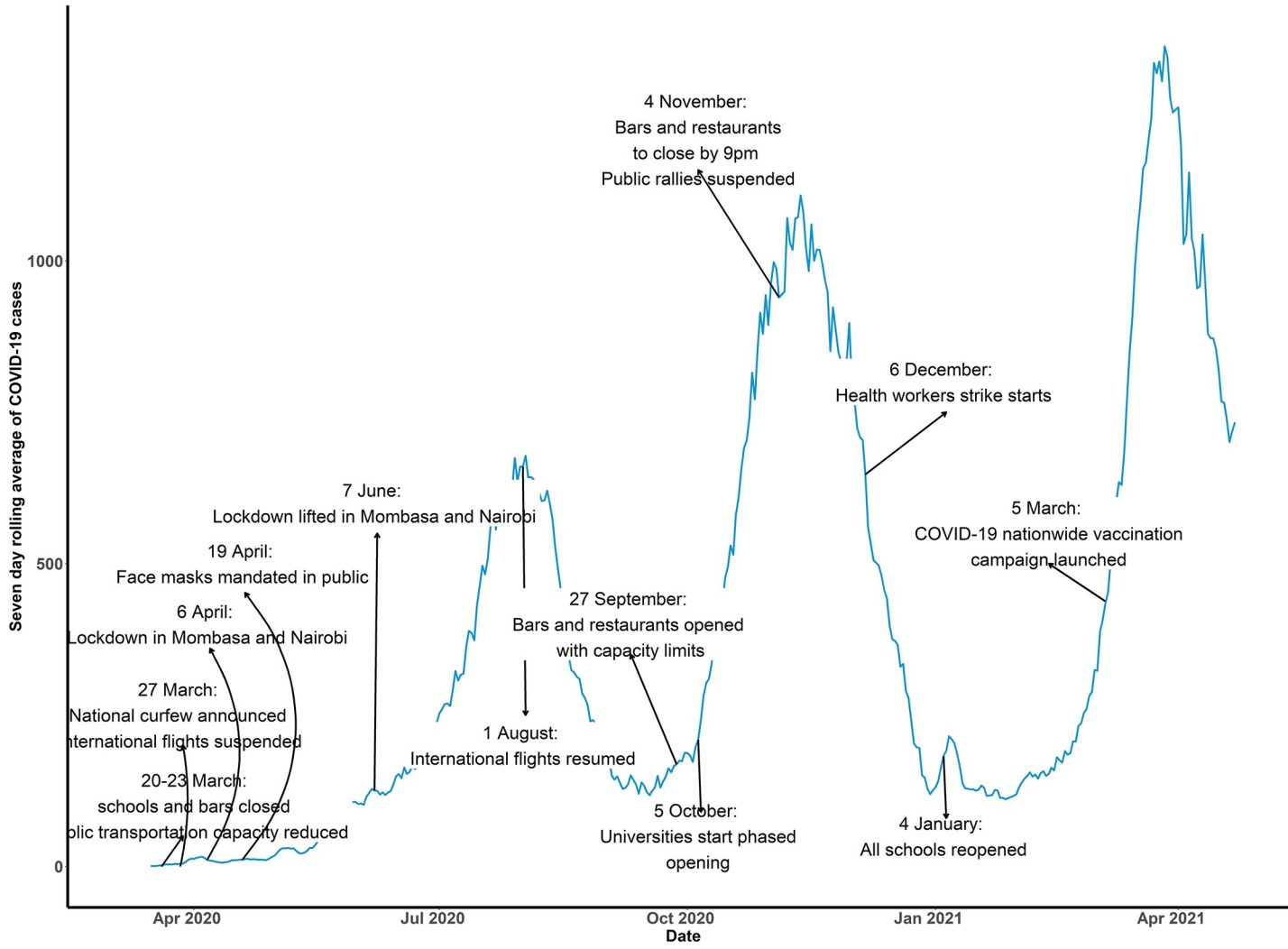

**Fig 1. COVID-19 control measures timeline during the period of study.**

## Data

**Selection of hospitals.** Hospitals were selected based on service availability. Data from varied databases on service availability were consulted. These include; Service Availability and Readiness Assessment (SARAM) [12], Emergency Medicine Foundation of Kenya (EMFK) [13], Harmonised Health Facility Assessment (HHFA) [14], District Health Information System (DHIS2) [15] and the Master Facility List (MFL) [16]. A single database of facilities was then created using a combination of the data sources. Five common indicators used across different countries to delineate hospitals were used. These were availability of operating theatres, oxygen, blood transfusion, radiology and critical new-born care. Hospitals identified to have the five indicators were then matched with the number of inpatient beds in an iterative process to confirm the provision of inpatient services and to validate the use of inpatient bed numbers in the definition of hospitals. Different iterations were then conducted based on the number of admissions and service availability to identify facilities that were either likely to have been missed by all of the service availability assessment considered or may be classified as lower-level facilities but attract on average 5,408 admissions annually and therefore warrant upgrading to hospital status. A total of 201 hospitals fitted this criterion and were selected. The indicators used to select the hospitals is presented in Table 1 in S1 File. Three more hospitals, which are approved internship centres by the Kenya Medical Practitioners and Dentists Council [17] were added to the list totalling to 204 hospitals. The distribution of the 204 hospitals is represented in Fig 1 & Table 2 in S1 File and the definition of the indicators used in this study presented in Table 3 in S1 File.

**Extraction of data.** Monthly data on inpatient admissions and deliveries in five patient groups (maternity, new-born units, paediatric, adult medical and surgical wards) were obtained from the District Health Information System (DHIS2), an open-source software platform for data reporting by most health facilities in the country. Data were extracted for the 204 hospitals, which are known to offer Comprehensive Emergency Obstetric and Newborn Care (CEmONC) services for the period January 2018 to March 2021, a total of 39 months. Adult medical data would be expected to include any admissions to general, high-dependency or intensive care wards although few hospitals will have these higher levels of care. We chose 2018 as a starting point for the data extraction because of prolonged health care workers' strikes in 2017 which affected health services provision [18] and consequently reporting. For each hospital, we extracted data on administrative units, level of hospital (Level 3: comprehensive primary health care facility, level 4: primary referral hospitals, Level 5: secondary referral hospitals and Level 6: national teaching and referral hospitals), whether a hospital is privately or publicly owned (ownership) and whether the hospital belongs to a clinical information network (CIN). The CIN is a collaboration of selected hospitals, policy makers and health researchers aimed at improving information available on the quality of inpatient paediatric care in Kenya [19–21]. A few hospitals reported data into an alternative parallel database (the hospital administrative database) and these data were merged where appropriate.

## Statistical analysis

**Handling missing data.** To adjust for incompleteness in reporting during the period of study, multiple imputation (MI) was performed [22–24]. Missing monthly values were imputed using a mixed effects model in a joint modelling framework using the *jomo* package in R and obtained 100 imputations for each indicator [25, 26]. We assumed the data followed as missing at random (MAR) pattern. The imputed datasets were analysed individually, and the estimates pooled using Rubin's Rules. Each inpatient indicator was used as an outcome in the imputation model. Ownership (public or private), whether hospital belongs to a clinical

information network (CIN), time (month and year) and COVID-19 binary indicator (0 – months before pandemic and 1 –months post pandemic) were used as covariates with the hospital ID as a clustering variable. MI was performed for hospitals with more than 30% of months reported (at least 11/39 months reported) to reduce uncertainty in imputed values and promote generalizability of the estimates. This strategy was supported by results from a simulation study we carried out, indicating MI performance and efficiency was best when imputing data for hospitals that reached this threshold of months reported. The imputation model specification is outlined in Handling missing data in S4 File. The number of hospitals included in analyses based on this strategy varied for each ward as summarised in the results with further information presented in Table 4 in S1 File.

### Interrupted time series analysis

**Exploratory analyses.**   Admissions and deliveries were aggregated monthly across all reporting hospitals and stratified by hospital ownership. Temporal trends were plotted separately for private and public hospitals to identify significant changes in health services and Statistical Process Control (SPC) charts were produced using 2018–2019 monthly averages as baseline to detect significant shifts in monthly admissions in 2020–2021 [27].

We also carried out multiple change point analysis [28, 29] to identify inpatient indicators that were affected by the national healthcare workers strike between December 2020 –February 2021. The assumption was that the strike affected all the public hospitals since we couldn't obtain a database that tracks strikes in hospitals nationally.

**Segmented regression.**   We conducted interrupted-time series analysis using monthly number of admissions by ward and deliveries as outcomes. Analysis was stratified by hospital type. For caesarean sections and admissions in NBU where we had an appropriate denominator (total deliveries and live births respectively), they were used as offset variable to convert them into a rate and adjust for any potential changes in utilization over time [30, 31]. We therefore report caesarean section rates and NBU admission rates. The period running from January 2018 through March 2020 was defined as pre-COVID-19 and April 2020 to March 2021 as post-COVID-19. For each indicator, a segmented regression model was fitted [30, 32]. The following equation specifies the model [32];

$$Y_t = \beta_0 + \beta_1 * time_t + \beta_2 * COVID19_t + \beta_3 * time\ after\ COVID19_t + e_t$$

Where, $Y_t$ is the number of admissions in month $t$; $time$ is a continuous indicator of time in months from January 2018; $COVID19$ is an indicator of time $t$ occurring before ($COVID19 = 0$) or after ($COVID19 = 1$) the outbreak, which was implemented at April 2021 in the series, where COVID-19 measures were most stringent in Kenya [33]; and $time\ after\ COVID19$ is a continuous variable counting the number of months after COVID-19 at time $t$. In the model, $\beta_0$ estimates the baseline level of admissions at time zero; $\beta_1$ estimates the change in the monthly number of deliveries before the intervention (the pre-existing trend); $\beta_2$ estimates the level change in monthly number of admissions immediately after COVID-19 outbreak, which is from the end of the preceding segment; $\beta_3$ estimates the change in the trend of monthly admissions after COVID-19, compared with the pre-existing trend. The random variability at time $t$, which is not explained in the model is represented by the error term $e_t$. A level change (immediate COVID-19 effect) and change in trend after COVID-19 were hypothesised [30].

A generalized linear model was applied, assuming a Poisson distribution. Since the count data was over dispersed, and after evaluating several models, we fitted the negative binomial model to account for overdispersion [30, 34, 35]. Model checking was conducted for

autocorrelation using the Durbin-Watson statistic and autoregressive moving average (ARMA) models were fitted for indicators with serial autocorrelation [36–38]. For indicators with significant autocorrelation, various combinations of the ARMA parameters (order of the autoregressive and moving average polynomials) were tested and the model resulting to lower Akaike Information Criterion (AIC) selected. Seasonality was adjusted using Fourier terms [39]. Results were pooled across the multiple imputed datasets using Rubin's rules [40]. The negative binomial segmented ITS model, which accounted for seasonality, autocorrelation and overdispersion was the best model when compared with negative binomial models that were not adjusted.

Sensitivity analyses were performed for indicators affected by the national health care workers strike by excluding months affected and comparing the estimates with those from data that included the strike period. In addition, we anticipated differences in reporting across hospitals. It is possible that the impact of the pandemic across hospitals was variable, for instance some hospitals were assigned as isolation centres or closed for some time. Also, the magnitude of the epidemic in some counties was higher. Therefore, in counties with higher case burden, some hospital staff were re-assigned or had to go for quarantine if exposed or isolation if they contracted COVID-19, leading to human resources shortages. The intraclass correlation coefficients across hospitals for each indicator is provided in Table 3 in S3 File. Therefore, a Poisson generalized estimating equations (GEE) was fitted for each indicator to test the effect of varying model assumptions, such as accounting for differences in hospitals, on the estimates [41].

Statistical significance was defined as p-values < 0.05. All analyses were performed using R (version 3.6.3).

## Ethics approval and consent to participate

The manuscript does not contain any individual person's data. We extracted aggregated monthly data at hospital level and not at patient level.

## Results

Out of the 204 hospitals evaluated in this study, 109 were public and 95 were private. The number and percentage of hospitals reporting at least 30% of months and were included in the analysis are presented in Table 3 in S1 File. In the sample of 109 public hospitals, all hospitals reporting at least a month were analysed for total deliveries and live births (108/108), caesarean sections (106/107), admissions in medical (102/104), admissions in surgical (48/57), admissions in paediatrics (100/106) and admissions in the NBU (61/73). Similarly, all the private hospitals reporting at least a month were included in the analysis for total deliveries and live births (95/95). While caesarean sections (93/94), admissions in medical (89/92), admissions in surgical (67/82), admissions in paediatrics (83/91) and admissions in NBU (32/60) in the sample of private hospitals.

SPC charts illustrate that for both public and private hospitals, no reported values fell outside the 3SD mark in the first three months of 2020 (pre-COVID) (Fig 2). In public hospitals, monthly admissions fell below the 3SD mark in the paediatrics, adult medical and surgical wards starting in April 2020 through June 2020. This substantial drop remained until November 2020 for admissions in the medical and paediatrics wards, while admissions in the surgical ward started returning to normalcy during this period. There were similar falls in private hospitals, with admissions in medical, paediatric and surgical wards below the 3SD threshold between April 2020 and August 2020, with slight recoveries for admissions in the surgical ward within this period. In this sample of private hospitals, although admissions in the medical and surgical ward started to recover starting September 2020 to the end of the study,

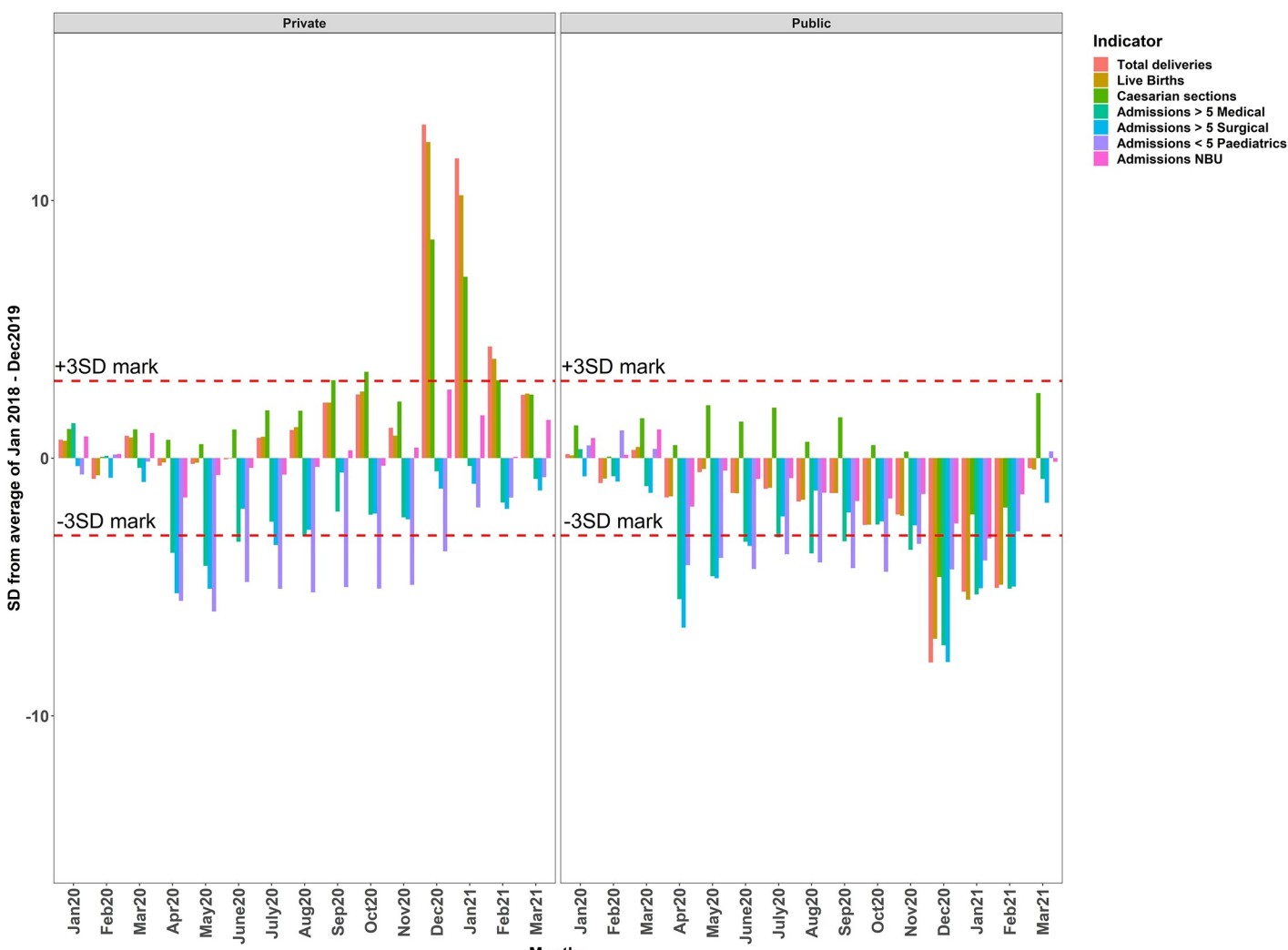

**Fig 2. Statistical Process Control chart for both public and private hospitals.** Horizontal dashed lines represent the 3-standard deviation mark.

reductions in admissions in the paediatrics ward remained below the 3SD mark up to December 2020, where they started going back to expected levels. Numbers of deliveries and live births in public hospitals also fell starting in April 2020 but these were less substantial than for other admissions and the sample of private hospitals utilisation of the maternity ward services began to rise through July 2020 to November 2020. In December 2020, during the nationwide healthcare workers strike, there was a drastic drop for all indicators in public hospitals and the highest increase in admissions for private hospitals (Fig 2). These changes continued to February 2021, with services beginning to go back to historical levels in March 2021.

## Interrupted timeseries results

From Table 1, before COVID-19, the trend was generally positive across all the indicators in both public and private hospitals, showing an increase in admissions and deliveries overtime.

There was an immediate drop in admissions post-COVID-19 for most inpatient indicators. In public hospitals, there was a 11% (IRR = 0.89 CI: 0.80–0.99), 29% (IRR = 0.71 CI: 0.63–0.79), 59% (IRR = 0.42 CI: 0.38–0.45) and 25% (IRR = 0.75 CI: 0.64–0.87) reductions in live

**Table 1.  Interrupted time series analysis results showing incidence rate ratios (IRR) and relative risk (RR)for COVID-19 intervention, time and trend alongside 95% confidence intervals.**

| | | Total deliveries | | | Live births | | | Caesarean sections rate | | | Admissions > 5 Medical | | |
|---|---|---|---|---|---|---|---|---|---|---|---|---|---|
| Ownership | | IRR | 95%CI | P-value | IRR | 95%CI | P-value | RR | 95%CI | P-value | IRR* | 95%CI* | P-value* |
| Public | COVID-19 | 0.89 | (0.79–1.02) | 0.08 | 0.89 | (0.80–0.99) | 0.04 | 1.10 | (1.06–1.14) | <0.01 | 0.71 | (0.63–0.79) | <0.01 |
| | Time | 1.00 | (0.99–1.01) | 0.67 | 1.00 | (0.99–1.01) | 0.42 | 1.01 | (1.00–1.01) | <0.01 | 1.00 | (0.99–1.00) | 0.57 |
| | Trend | 0.99 | (0.97–1.00) | 0.11 | 0.99 | (0.97–1.00) | 0.07 | 1.00 | 0.99–1.00) | 0.14 | 1.01 | (0.99–1.03) | 0.10 |
| Private | COVID-19 | 0.91 | (0.81–1.03) | 0.13 | 0.91 | (0.81–1.03) | 0.12 | 1.03 | (0.98–1.09) | 0.25 | 0.66 | (0.60–0.72) | <0.01 |
| | Time | 1.00 | (0.99–1.01) | 0.06 | 1.00 | (0.99–1.01) | 0.03 | 1.00 | (1.00–1.01) | 0.02 | 1.00 | (1.00–1.01) | 0.01 |
| | Trend | 1.03 | (1.01–1.04) | <0.01 | 1.03 | (1.01–1.04) | <0.01 | 0.99 | (0.98–0.99) | 0.01 | 1.03 | (1.01–1.04) | <0.01 |

| | | Admissions < 5 Paediatrics | | | Admissions > 5 Surgical | | | NBU Admissions rate | | |
|---|---|---|---|---|---|---|---|---|---|---|
| Ownership | | IRR* | 95%CI* | P-value* | IRR* | 95%CI* | P-value* | RR | 95%CI | P-value |
| Public | COVID-19 | 0.41 | (0.38–0.45) | <0.01 | 0.75 | (0.64–0.87) | <0.01 | 0.92 | (0.83–1.01) | 0.09 |
| | Time | 1.00 | (1.00–1.01) | <0.01 | 1.00 | (0.99–1.00) | 0.46 | 1.00 | (1.00–1.01) | 0.02 |
| | Trend | 1.04 | (1.03–1.05) | <0.01 | 1.01 | (0.99–1.03) | 0.19 | 1.01 | (1.00–1.02) | 0.04 |
| Private | COVID-19 | 0.38 | (0.33–0.45) | <0.01 | 0.72 | (0.64–0.81) | <0.01 | 0.87 | (0.72–1.04) | 0.13 |
| | Time | 1.00 | (0.99–1.01) | 0.38 | 1.00 | (0.99–1.00) | 0.82 | 1.00 | (0.99–1.01) | 0.21 |
| | Trend | 1.07 | (1.05–1.08) | <0.01 | 1.02 | (1.01–1.04) | <0.01 | 0.99 | (0.97–1.02) | 0.48 |

*Estimates from ARMA models. The ARMA (p,q) parameters for admissions in medical, surgical and paediatrics are (2,0), (1,0) and (2,0) respectively.

births, over-fives admissions in medical, under-fives admissions in paediatrics and over-fives admissions in surgical respectively. Over-fives in medical, under-fives paediatrics and over-fives surgical admissions declined by 34% (IRR = 0.66 CI: 0.60–0.72), 62% (IRR = 0.38 CI: 0.33–0.45) and 28% (IRR = 0.72 CI: 0.64–0.81) respectively in private hospitals. Caesarean section rate increased by 10% (RR = 1.10 CI: 1.06–1.14) in public hospitals. Although monthly admissions in the postCOVID-19 period (May 2020 onwards) experienced some recovery across all the indicators except for NBU admissions rate in the sample of private hospitals, the recovery was statistically significant for admissions in the paediatrics ward in the sample of public hospitals. There was significant autocorrelation for admissions in medical, surgical and paediatrics wards and estimates presented are produced from the ARMA models. The ITS fitted models are presented in Fig 3.

## Sensitivity analyses

Multiple change point analysis showed the national strike between December 2020 –February 2021 coincided with significant reductions in live-births and caesarean sections in the sample of public hospitals. Conversely, during this strike period, total deliveries, live-births and admissions in the paediatrics ward increased sharply in private hospitals (Figs 2 and 3 in S2 File). Excluding data for the three months when strike was ongoing showed the estimates for the post-COVID-19 trend were sensitive to changes in services during the strike for total deliveries, live-births and caesarean section rates in the sample of private hospitals and for surgical and NBU admissions in public hospitals (Tables 1 & 2 in S3 File). The estimates obtained from the Poisson generalized estimating equations (GEE) were not different from the primary model estimates hence robustness of the estimates (Table 4 in S3 File).

## Discussion

In this study, the unintended impact of COVID-19 control measures on inpatient admissions for a large sample of private and public Kenyan hospitals using monthly data reported to the

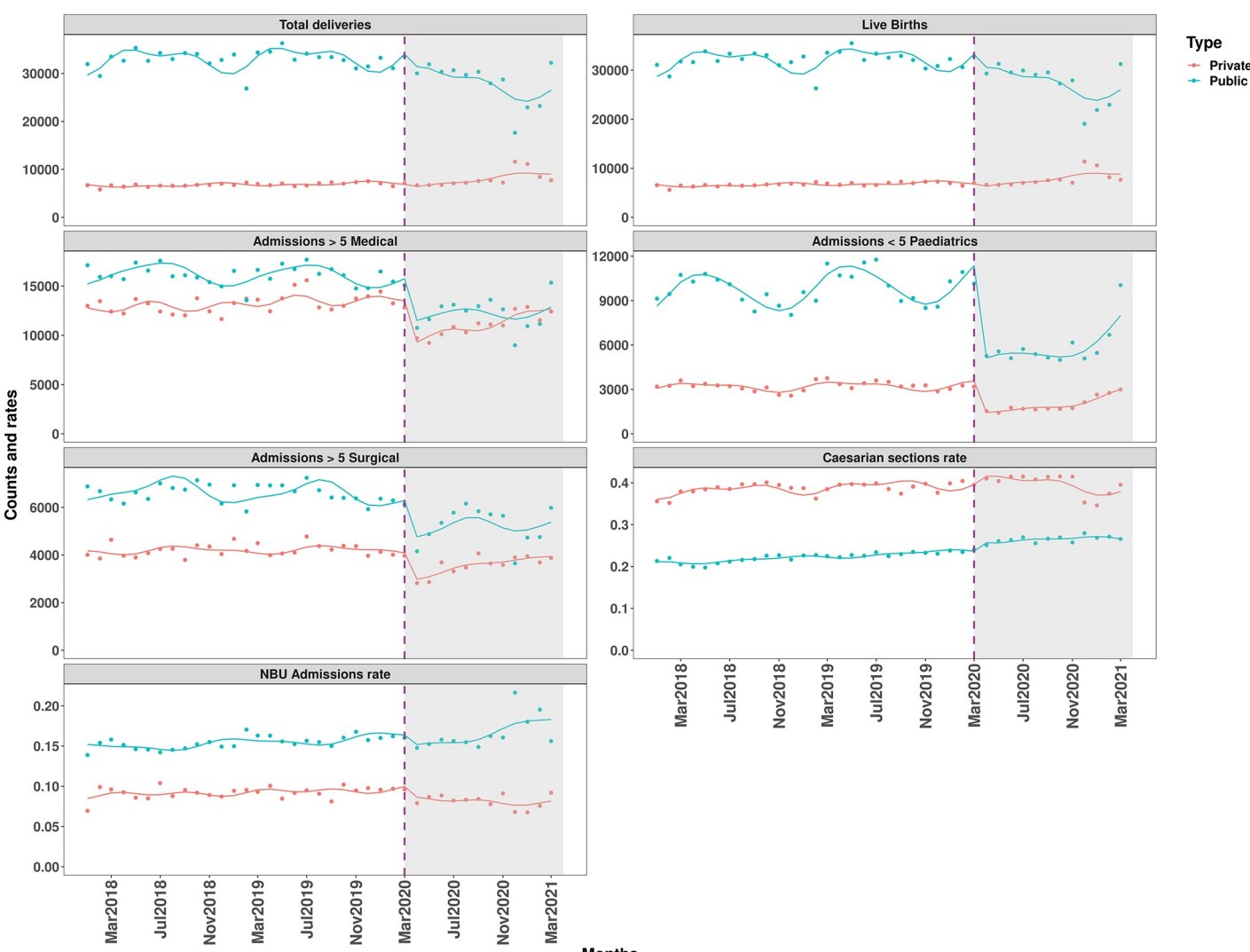

**Fig 3. Fitted lines of segmented regression models for all indicators and by hospital ownership.** Vertical lines represent the month (March 2020) COVID-19 was announced in Kenya and as a pandemic by the WHO.

national information system from January 2018 to March 2021 was evaluated. Utilization of inpatient services was rising slowly prior to March 2020 in these non-tertiary hospitals but declined sharply after this co-incidental with the first introduction of national measures to restrict population mixing in the initial wave of Kenya's epidemic. Paediatric admissions fell especially sharply in both public and private sectors. So too, although to a lesser degree, did adult surgical and medical admissions in these hospitals that would typically not have any significant high-dependency or intensive care capacity [42]. In the public sector admissions for these three older groups, including adult medical and surgical ward admissions, remained lower than was typical for the following 12 months during what appeared to be Kenya's first and second wave of SARS-CoV-2 infections. During this period a national health workers' strike between in December 2020 and February 2021 also likely reduced admissions for inpatient care even though our analysis shows that this didn't significantly affect all the indicators uniformly. In the private sector admission volumes for these three groups slowly rose again during this 12 months' period but did not exceed pre-COVID volumes.

Interestingly, total deliveries and total live births in the public sector, that we would obviously expect to be closely linked, were less dramatically affected at the start of Kenya's epidemic. However, public sector utilisation of maternity services declined slowly over the 12 months' post-COVID period with a nadir during the December 2020 health workers' strike. The decline in public sector utilisation of hospital maternity services was associated with a small but significant increase in the proportion of women having a caesarean delivery. Caesarean delivery rates fell during the health workers' strike (Fig 3). Over the same post-COVID period the proportion of live births admitted to newborn units also slowly increased. Taken together these findings may suggest that while public sector delivery volumes were generally declining. The number of women delivering in hospital with complications of pregnancy, and consequently of newborns with problems at birth, did not decline as much. In the private sector, there was no decline in delivery volumes of live births and in fact a slow increase in these volumes over time. The much higher proportion of women in the private sector (35–40%) delivering by caesarean section is also worthy of note while the proportion of live births being admitted to NBU in private hospitals is much lower than in the public sector.

The discrepancy between paediatric, adult medical and adult surgical admissions' indicators for which volumes fell sharply and use of maternity services for delivery is worthy of note and somewhat unexpected. The majority of hospitals for which we have data are non-tertiary and at Level 4 and 5 of the Kenyan health system and located in the major towns in counties and in urban and peri-urban settings of Kenya's cities. It is therefore possible that urban families prefer to use hospitals for maternity services rather than deliver at home or in smaller clinics. Use of public hospitals may also have been sustained by the policy of free maternity services in the public sector policy put in place by the Kenyan government in 2013. Despite this there was a suggestion that as deliveries declined in the public sector they increased in the private sector, something likely exacerbated by the public sector health workers' strike in December 2020.

The significant reductions in over-fives admissions in both medical and surgical wards we hypothesize are likely linked to the restrictions in movements, dusk-to-dawn curfews, lack of or high cost of transport that were the result of COVID containment strategies. It is also possible, as public hospital care is not free for these populations, that the loss of income experienced by many in the post-COVID period prevented care seeking. The cases of the particularly large drop in under-fives admissions from April 2020 are not known. We can speculate that, in addition to the factors influencing utilisation of hospitals by adults, parent's might fear taking their children to hospitals to limit their own and their child's exposure to contracting COVID-19 infection [43]. It is also possible, that rates of illness were reduced because families paid greater attention to hygiene (e.g. hand-washing) [44], because of social distancing and because schools were closed limiting transmission of many respiratory illnesses in households [45] as respiratory illness is responsible for a large proportion of paediatric admissions in Kenya [46].

The patterns of decline in admissions post-COVID-19 reflect those observed in previous hospital and population-based studies [41, 47–50]. A study in the US observed dramatic reductions in hospital admissions after onset of COVID-19 [47]. Significant reductions in childcare services were also reported in South Africa following lockdown measures [41]. Although no studies have quantified effects of the pandemic on surgeries in Kenya, early estimates had revealed reductions in surgical case volumes in low- and middle-income countries [51]. Additionally, deficit in the provision of personal protective equipment, surgical masks, gloves, N95 masks, and shoe covers were reported to worry surgeons in a cross-sectional study of neurosurgeons across African countries, which might affect provision and consequently utilization of these services [52].

## Strengths of the study

This study provides an understanding of the current situation on utilization of inpatient admissions in a large sample of hospitals in Kenya in the face of the COVID-19 pandemic. At a time when most attention is on restrictive measures to combat the pandemic, the unintended effects of these policies might be overlooked and measuring the effects should inform ongoing policy discussions that seek to balance COVID prevention and ensuring people can access care. This study adjusted for potential confounders in assessing changes overtime. Firstly, we adjusted for missing data in DHIS2, which is quite common in routine data but not always done [49, 53]. This may be important as non-reporting may be compounded during COVID-19 if Health Records and Information Officers work at reduced capacity during the epidemic. Additionally, the WHO in a recent guide on using routine data to monitor the effects of COVID-19 recommended adjusting for low reporting rates [54]. More generally, missing data were often encountered and to gain an accurate and rapid picture of the effects of COVID policies in countries such as Kenya more needs to be done to improve DHIS2 data quality. This will likely require investment in better infrastructure, supervisory support, formal data quality assurance and human resources [55, 56]. A second strength was use of data from a reasonably large number of hospitals from a broad geographic region covering all the counties across the country. Thirdly, compared to previous studies [49, 57], we have used a prolonged pre-COVID-19 period and a full 12 months post-COVID-19 period. This provided enough data points to measure the post-COVID-19 trend with precision and also allow adjustments for seasonal trends. Finally, we evaluated the potential effect of a second interruption to services, the health care workers' strike, and used sensitivity analysis to account for uncertainty in estimates due to hospital level variability.

## Limitations

We recognize some limitations of the study. There is poor reporting into DHIS2 of private hospitals, especially those offering the equivalent of regional or tertiary level services. This reduces the number or private hospitals in our sample and tends to exclude those with more advanced patient care facilities. We also had to exclude some tertiary and more specialised centres from the public sector because these too reported less than 30% of months data. Our finding that adult medical admissions were reduced may therefore be misleading if these larger hospitals experienced a major rise in admissions if adults with severe COVID-19 sought care at these sites preferentially. Importantly, we sought to analyse data on mortality of the admissions analysed in this report. However, DHIS2 as implemented in Kenya makes this extremely challenging as hospitals are unable to record zero deaths in a month, efforts to enter a zero-value result in a missing value in the actual database. Accounting for this will need more complex imputation procedures which are being explored. Additionally, given the nature of the pandemic there are no potential controls for this study and so while it seems entirely plausible that the effects, we describe are a consequence of the introduction of COVID-19 control measures it remains possible there are other unknown explanations. We also note that the analysis was restricted to events occurring within hospitals and doesn't account for those occurring in primary health facilities or outside the health system. We evaluated the impact of health worker strike on the analysis assuming a uniform impact across all hospitals due to lack of a database that tracks strikes within hospitals nationally, however the length of the strike might have been shorter for some hospitals. Lastly, we didn't carry out sensitivity analysis, but this is work we are considering in the future to assess any departures from the MAR assumption.

## Conclusion

In a large sample of public hospitals significant drops were observed in monthly volumes of live births (11%), over-fives admissions for medical (29%) and surgical care (25%) with especially high declines in admissions in the under 5 age group (59%). Similarly, substantial declines were apparent in private hospitals, where significant reductions in admissions to the medical (34%), surgical (28%) and paediatrics (62%) wards were observed. However, maternity ward indicators and NBU admission rates were not affected in this sample of private hospitals. The declines have been sustained and recent data suggests a reversal in trends with services appearing to be going back to the historical levels starting March 2021.

## Supporting information

**S1 File. Distribution of hospitals and number of health facilities analysed for each indicator including health facilities excluded for not reporting any month and those with less than 30% of months reported.** The definition of the indicators analysed is outlined.
(DOCX)

**S2 File. Patterns of missing data across all the health facilities analysed including multiple change point analysis plots.**
(DOCX)

**S3 File. Sensitivity analysis estimates.**
(DOCX)

**S4 File. Multiple imputation methodology.**
(DOCX)

## Author Contributions

**Conceptualization:** Steven Wambua, Mike English, Emelda A. Okiro.

**Data curation:** Steven Wambua, Lucas Malla, George Mbevi, Amen-Patrick Nwosu, Timothy Tuti, Chris Paton, Samuel Cheburet, Ayub Manya, Mike English, Emelda A. Okiro.

**Formal analysis:** Steven Wambua, Lucas Malla, Mike English, Emelda A. Okiro.

**Funding acquisition:** Mike English, Emelda A. Okiro.

**Investigation:** Steven Wambua, Lucas Malla, George Mbevi, Amen-Patrick Nwosu, Timothy Tuti, Chris Paton, Samuel Cheburet, Ayub Manya, Mike English, Emelda A. Okiro.

**Methodology:** Steven Wambua, Lucas Malla, George Mbevi, Amen-Patrick Nwosu, Timothy Tuti, Chris Paton, Samuel Cheburet, Ayub Manya, Mike English, Emelda A. Okiro.

**Project administration:** Steven Wambua, Mike English, Emelda A. Okiro.

**Resources:** Mike English, Emelda A. Okiro.

**Software:** Steven Wambua, Lucas Malla.

**Supervision:** Mike English, Emelda A. Okiro.

**Validation:** Steven Wambua, Lucas Malla, George Mbevi, Amen-Patrick Nwosu, Timothy Tuti, Chris Paton, Samuel Cheburet, Ayub Manya, Mike English, Emelda A. Okiro.

**Visualization:** Steven Wambua, Lucas Malla, George Mbevi, Amen-Patrick Nwosu, Timothy Tuti, Samuel Cheburet, Mike English, Emelda A. Okiro.

**Writing – original draft:** Steven Wambua.

**Writing – review & editing:** Lucas Malla, George Mbevi, Amen-Patrick Nwosu, Timothy Tuti, Chris Paton, Samuel Cheburet, Ayub Manya, Mike English, Emelda A. Okiro.

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
