## [Decision Letter · Decision Letter 0]

2 Sep 2021

 PGPH-D-21-00130 The indirect impact of COVID-19 pandemic on inpatient admissions in 204 Kenyan hospitals: An interrupted time series analysis PLOS Global Public Health

Dear Dr. Wambua,

Thank you for submitting your manuscript to PLOS Global Public Health. After careful consideration, we feel that it has merit but does not fully meet PLOS Global Public Health’s publication criteria as it currently stands. Therefore, we invite you to submit a revised version of the manuscript that addresses the points raised during the review process.

We look forward to receiving your revised manuscript.

Kind regards,

Jake Michael Pry, PhD, MPH

Guest Editor

Journal Requirements:

Additional Editor Comments (if provided):

Good Day,

Thank you for your submission to PLOS Global Public Health (PGPH-D-21-00130). We have reviewed your manuscript and request minor revisions. In addition to comments from reviewers we would like additional detail for the following items.

- Selection criteria for the hospitals/care facilities to include is quite important and would like more explicit language, if possible, in and around line 119 where a "considerable number" of inpatients justified inclusion.

- Please provide additional detail/justification, if possible, for why the national healthcare worker strike was limited to the public sector facilities.

Many thanks, again, for your submission and we look forward to reviewing revisions.

Reviewers' comments:

Reviewer's Responses to Questions

**Comments to the Author**

1. Does this manuscript meet PLOS Global Public Health’s publication criteria? Is the manuscript technically sound, and do the data support the conclusions? The manuscript must describe methodologically and ethically rigorous research with conclusions that are appropriately drawn based on the data presented.

Reviewer #1: Yes

Reviewer #2: Yes

2. Has the statistical analysis been performed appropriately and rigorously?

Reviewer #1: Yes

Reviewer #2: No

3. Have the authors made all data underlying the findings in their manuscript fully available (please refer to the Data Availability Statement at the start of the manuscript PDF file)?

Reviewer #1: Yes

Reviewer #2: Yes

4. Is the manuscript presented in an intelligible fashion and written in standard English?

Reviewer #1: Yes

Reviewer #2: Yes

5. Review Comments to the Author

Reviewer #1: This is an excellent paper on the indirect impact of COVID-19 pandemic on inpatient admissions in 204 Kenyan

hospitals. The paper is well-written, and methodologically sound. The authors did not only use the DHIS2 data sets that have some weakness, but took care of the robustness and validity of the data by using other hospital data systems.

The stats are well elaborated; the result section provides interesting data of 5 key indicators of hospital admissions, and leads to important observations that are well supported and discussed in the discussion section. Limitations are described as well.

The authors have observed public and private hospitals and described delivery trends in covid times, taking into account also the effects of the health care workers strike- indicating the resilience of the maternal health care systems in Kenya. Obviously, more difficult to measure is the number of births at home or with TBAs, as the birth registration systems are incomplete in many parts of the country.

In conclusion: great work!

Reviewer #2: Line 54: the authors write: “The spread of the pandemic has placed unpresented challenge on health systems.” I presume they meant unprecedented

What was the impact of the lockdowns? One could argue that the lockdown restricting mobility was the direct course as opposed to health facility stigma as vectors of transmission. This paper will improve dramatically if the authors include mobility data e.g. facebook/google mobility data for social good.

Line 145-147: It is unclear how the authors implemented multiple imputation for missing data. Reviewers and the readership may benefit from a complete presentation of the algorithm used in the imputation process as supplementary material. What assumptions were made and whether sensitivity analyses for the missing data mechanism were performed. I am especially keen to see how the joint model was formulated /specified.

The authors mention the covariates used in the imputation model, but no information on which variables/indicators were imputed. The number imputations is also missing.

Line 188: The authors say, “A generalized linear model was applied, assuming a Poisson distribution. We fitted both Poisson and negative binomial models to account for overdispersion (30-32)”. There is no reason why the authors are fitting both models unless it is at the point of model selection in which case only the optimum model results should be presented in the main results. Was the data over dispersed? Given the volume of the data, does the Poisson model approximate the normal distribution?

Line 197 “a hospital level mixed-effects model using generalized estimating equations (GEE)” is incorrect. This is because mixed effects and marginal GEE are two distinct model families and parameter estimates one model family cannot be used to assess robustness of the other more so when dealing with non-continuous outcomes.

Hence Line 257: “The estimates obtained from hospital level generalized estimating equations (GEE) were not different from the primary model estimates hence robustness of the estimates.” could be misleading for the reason above

Sensitivity analyses with the GEE model was performed to assess the potential variability by hospital/health facility. Please indicate the distribution of the error term (Normal, Poisson or negative binomial). Were some hospitals more affected by the pandemic compared to other? Explain the variability observed

I would also expect the so see parameter estimates for the variance between hospitals

The authors need to define RR when it is first used. Count models output the incidence rate ratios (IRR) and not the risk ratios (RR).

Caesarean section rate and NBU the authors considered proportions and but fitted Poisson and negative binomial. why didnt they consider beta regression model ?

Lines 231-232: “The negative binomial segmented ITS model, which accounted for seasonality, autocorrelation and overdispersion was the best model when compared with Poisson and negative binomial models that were not adjusted”. However, parameter estimates for the ARMA(p,q) part of the best fitting model are missing in the results section.

Line 283. sentence starting with “ the..

6. PLOS authors have the option to publish the peer review history of their article (what does this mean?). If published, this will include your full peer review and any attached files.

**Do you want your identity to be public for this peer review?** For information about this choice, including consent withdrawal, please see our Privacy Policy.

Reviewer #1: **Yes: **Marleen Temmerman

Reviewer #2: No

---

## [Editor Report · Decision Letter 1]

25 Oct 2021

The indirect impact of COVID-19 pandemic on inpatient admissions in 204 Kenyan hospitals: An interrupted time series analysis

PGPH-D-21-00130R1

Dear Dr. Wambua,

We're pleased to inform you that your manuscript has been judged scientifically suitable for publication and will be formally accepted for publication once it meets all outstanding technical requirements.

Within one week, you'll receive an e-mail detailing the required amendments. When these have been addressed, you'll receive a formal acceptance letter and your manuscript will be scheduled for publication.

An invoice for payment will follow shortly after the formal acceptance. To ensure an efficient process, please log into Editorial Manager at https://www.editorialmanager.com/pgph/ click the 'Update My Information' link at the top of the page, and double check that your user information is up-to-date. If you have any billing related questions, please contact our Author Billing department directly at authorbilling@plos.org.

Kind regards,

Jake Michael Pry, PhD, MPH

Guest Editor

Additional Editor Comments (optional):

Dear Dr. Wambua,

Thank you for your thoughtful revisions. We recommend that this revised draft be accepted for publication.

Many Thanks,

Jake M. Pry, PhD, MPH

Guest Editor

PLoS Global Public Health